# LEARNING TO REWRITE: GENERALIZED DETECTION OF LLM-GENERATED TEXT

## ABSTRACT

Large language models (LLMs) present significant risks when used to generate non-factual content and spread disinformation at scale. Detecting such LLM-generated content is crucial, yet current detectors often struggle to generalize in open-world contexts. We introduce **Learning2Rewrite**, a novel framework for detecting AI-generated text with exceptional generalization to unseen domains. Our method leverages the insight that LLMs inherently modify AI-generated content less than human-written text when tasked with rewriting. By training LLMs to minimize alterations on AI-generated inputs, we amplify this disparity, yielding a more distinguishable and generalizable edit distance across diverse text distributions. Extensive experiments on data from 21 independent domains and four major LLMs (GPT-3.5, GPT-4, Gemini, and Llama-3) demonstrate that our detector outperforms state-of-the-art detection methods by up to 23.04% in AUROC for in-distribution tests, 37.26% for out-of-distribution tests, and 48.66% under adversarial attacks. Our unique training objective ensures better generalizability compared to directly training for classification, when leveraging the same amount of learned parameters. Our findings suggest that reinforcing LLMs' inherent rewriting tendencies offers a robust and scalable solution for detecting AI-generated text.

## 1 INTRODUCTION

Large Language Models (LLMs) demonstrate exceptional capabilities across various tasks (Radford et al., 2019; Brown et al., 2020; Achiam et al., 2023; Touvron et al., 2023; Team et al., 2023; OpenAI, 2020). However, they can be misused for illegal or unethical activities, such as spreading misinformation (Chen & Shu, 2023), scaling spear phishing campaigns (Hazell, 2023), facilitating social engineering and manipulation of social media (Zhang et al., 2024), and generating propaganda (Pan et al., 2023). LLMs also facilitate academic dishonesty (Zellers et al., 2019; Mvondo et al., 2023), and training foundation models with generated content can lead to irreversible defects in resulting models (Shumailov et al., 2023). These issues highlight the urgent need for reliable algorithms to detect LLM-generated text.

Various methods for detecting generated text have been proposed (Solaiman et al., 2019; Fagni et al., 2021; Mitrović et al., 2023; Mitchell et al., 2023; Su et al., 2023; Liu et al., 2024; Bao et al., 2024; Mao et al., 2024). Most of these detectors employ pre-trained models, extracting hand-crafted features and heuristics, such as loss curvature (Bao et al., 2024) and rewriting distance (Mao et al., 2024), and apply thresholds to distinguish LLM from human data. However, these thresholds are highly domain-dependent, obfuscating the establishment of a universal detection standard.

In this paper, we present L2R (Learning to Rewrite), which trains an LLM to perform more edits when being asked to rewrite human-generated data and fewer edits when rewriting on LLM-generated data across a diverse set of domains. Unlike traditional detectors, which work well in-distribution (ID) but often struggle to generalize among out-of-distribution (OOD) domains (including adversarial attacks), our algorithm leverages

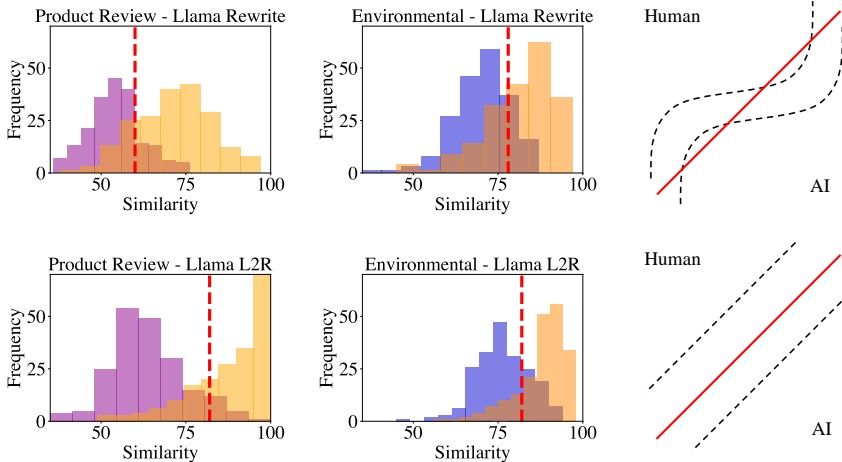

Figure 1: **Rewriting for LLM Text Detection**. The histograms on the left show the similarity between human and AI texts before and after fine-tuning a rewrite model on two domains. Purple and Yellow represent human and AI distributions for Product Review texts, while Blue and Orange represent those for Environmental texts. Initially, human and AI texts are inseparable by a single threshold (red line, above). After fine-tuning, the texts can be separated by this threshold (below). On the right, we conceptualize L2R's intuition by showing that the rugged decision boundary between human and AI texts, caused by varying data distributions across domains, can be better aligned and divided by a single threshold after fine-tuning.

the inherent tendency of LLMs to modify their own output less frequently, and maximizing its generalizability by focusing on learning a single rewriting threshold across diverse distributions. Figure 1 illustrates an example of how L2R learns to make LLM and human generated text more separable across domains, comparing with rewriting using a pre-trained model (Mao et al., 2024).

Visualizations and numerical results demonstrate that our targeted training objective enables LLMs to better capture the intricate structure of AI-generated content. To reflect the rapid advancements and real-world diversity of LLM-generated text, we in addition constructed a dataset spanning 21 domains (e.g., finance, entertainment, cuisine) using four different generator models. L2R surpasses the state-of-the-art detectors, achieving up to 19.56% higher AUROC ID and 35.10% higher OOD than Verma et al. (2024), 23.04% higher ID and 37.26% higher OOD than Bao et al. (2024), and 10.39% higher ID and 4.67% higher OOD than Mao et al. (2024). Comparing with fine-tuning a Llama-3 model for naive text classification, L2R has 51.35% higher AUROC OOD despite leveraging the same number of parameters. These results demonstrate that our training objective offers superior accuracy and generalizability. Furthermore, our method provides interpretability by highlighting the rewritten portions of the text. We will release our data, code, and models upon acceptance.

## 2 RELATED WORK

Various AI-generated text detectors have been proposed over the years. One set of detectors directly train a model on the input text (Solaiman et al., 2019; Fagni et al., 2021; Mitrović et al., 2023). These methods excel in their training domains but struggle under OOD evaluation (Uchendu et al., 2020; Pu et al., 2023), namely detection with text from different domains or unfamiliar models. The second set of detectors utilize the raw outputs, i.e., logits, from pre-trained LLMs to assign probability score for detection. GLTR (Gehrmann et al., 2019) utilizes statistical features like log probability, word rank, and entropy to

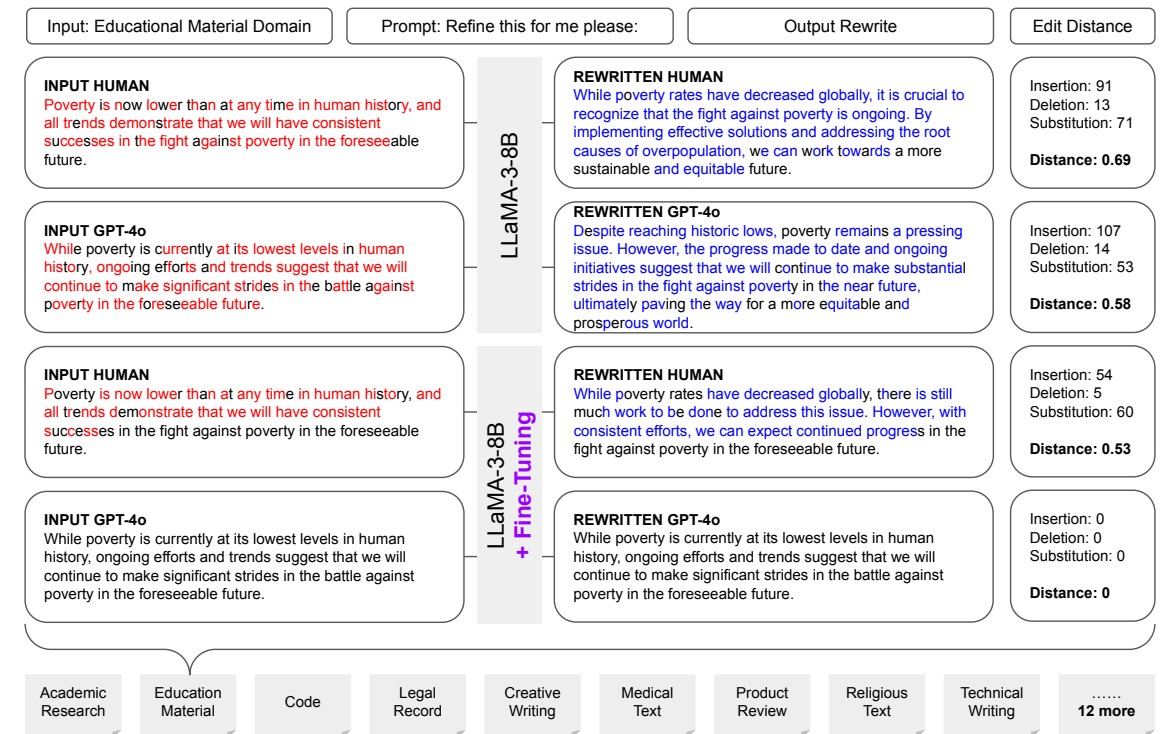

Figure 2: **Overview.** Deleted characters are marked in red, added characters are marked in blue, and unmodified characters are in **black**. We exploit the difference in rewriting distance between human and AI texts for classification. While the off-the-shelf Llama-3 model give different amount of rewrite for human and AI texts (above), rewrites from our fine-tuned model are much more separable (below).

assign score, Ghostbuster (Verma et al., 2024) utilizes log probability and unigram and bigram probability, DetectGPT (Mitchell et al., 2023) employs the delta in log probability of the input text after token perturbation to estimate AI likelihood, and Fast-DetectGPT (Bao et al., 2024) simplifies the process by exploiting conditional probability curvature. Ippolito et al. (2020) reveals that decoding strategies optimized for human-like text inadvertently introduce statistical artifacts that automated systems can detect with ease. These family of detectors all require raw output of an LLM in some way or the other, but the main target of detection, namely commercial LLMs, are not open-sourced, which potentially impose a barrier on their probability estimation. Lastly, RAIDAR (Mao et al., 2024) is a detection method based on the observation that LLMs, when prompted to rewrite a given text, tend to produce a greater number of rewrites for human-written text compared to AI-generated text. Despite the attempt on capturing rewrite edit distance as a domain-agnostic feature, the rewrite amount still varies across distributions, and the threshold of rewrite amount between human and AI texts learned on training domains does not generalize to OOD, which limits its full potential.

## 3 METHOD

### 3.1 REWRITING FOR LLM DETECTION

Rewriting input with LLM and then measuring the change proves to be a successful way to detect LLM-generated content. Given an held-out input text set $\mathbf{X}_{train}$ with LLM and human generated text, and its

corresponding label set $\mathbf{Y}_{train}$, an LLM $F(\cdot)$ is prompted to rewrite the input $\mathbf{x} \in \mathbf{X}_{train}$ using a prompt $\mathbf{p}$. The rewriting output is $F(\mathbf{p}, \mathbf{x})$. Particularly, the prompt $\mathbf{p}$ can be set to: `Refine this for me please`.

The edit distance between the input text and the rewritten output, $D(\mathbf{x}, F(\mathbf{p}, \mathbf{x}))$, is then computed for all $\mathbf{x} \in \mathbf{X}_{train}$. Mao et al. (2024) adopts the Levenshtein distance (Levenshtein et al., 1966), which is defined as the minimum number of insertions, deletions, or substitutions required to transform one text into the other. With the Levenshtein distance, a similarity score we use for classification is calculated based on:

$$D_k(\mathbf{x}, F(\mathbf{p}, \mathbf{x})) = 1 - \frac{\text{Levenshtein}(F(\mathbf{p}, \mathbf{x}), \mathbf{x})}{\max(len(F(\mathbf{p}, \mathbf{x})), len(\mathbf{x}))}. \tag{1}$$

Mao et al. (2024) trains a classifier, such as logistic regression or decision tree, to threshold the similarity scores and predict if it is written by an LLM. However, as shown in Figure 1, the threshold of rewriting with a vanilla LLM often varies from one domain to another, causing RAIDAR to fail to generalize to new domains.

### 3.2 Fine-Tuning the Rewrite Model

L2R works on the premise that human-written and AI generated text would cause a different amount of rewrites and a boundary can be drawn to separate both distributions. Thus we can finetune such a rewrite model $F'(\cdot)$, that gives as much rewrite as possible for human texts, while leaving the AI texts unmodified, demonstrated in Figure 2. Given some human text $\mathbf{x}_h \in \mathbf{X}_{train}$ and AI text $\mathbf{x}_{ai} \in \mathbf{X}_{train}$, our objective becomes:

$$\max\{D(\mathbf{x}_h, F'(\mathbf{p}, \mathbf{x}_h)) - D(\mathbf{x}_{ai}, F'(\mathbf{p}, \mathbf{x}_{ai}))\} \tag{2}$$

Since the edit distance is not differentiable, we use the cross-entropy loss $L(\cdot)$ assigned to the input $\mathbf{x}$ by $F'(\cdot)$ as a proxy to the edit distance. As a result, for each of input $\mathbf{x}$ with label $y = 1$ (AI) or 0 (human), our loss function becomes:

$$\min\{L(\mathbf{X}_{\text{train}}) \cdot y_{\text{train}}\}, \quad y_{\text{train}} = \begin{cases} 1 & \text{(AI)} \\ -1 & \text{(human)} \end{cases} \tag{3}$$

In this way, we flip the sign of the loss of the human texts. Since the overall loss would be minimized, this effectively encourages the rewrites to be different from human input and identical to the AI input.

### 3.3 Calibration Loss during Fine-Tuning

When fine-tuning the rewrite model on Equation 3, the rewrite model aims to maximize the edits on human-generated text and minimize the edits on LLM-generated texts. However, without posting regularization and constraint on the unbounded loss, the rewrite model takes the risk of being corrupted (e.g., verbose output for all rewrite and over-fitting with more edits on human-generated text rewrite) which we evaluated in §A.5.

Therefore, we propose a calibration loss, which prevents the over-fitting problem by imposing a threshold value $t$ on the absolute value of the loss on each given input. For human text $\mathbf{x}_h$, we apply gradient backpropagation only if the absolute loss $L(\mathbf{x}_h) < t$. For AI text $\mathbf{x}_{ai}$, we apply backpropagation only if $L(\mathbf{x}_{ai}) > t$. Otherwise, the gradient is set to 0. We show a pseudocode for the algorithm in 1.

Therefore, rather than minimizing the loss proxy, our objective becomes separating the distribution of human and AI rewrites to two ends of the threshold $t$. Concretely, this enables the model to only optimize against the hard examples, and leave those already correctly classified unchanged, so that we prevent overfitting. This is similar to DPO (Rafailov et al., 2023), where we fine-tune the rewrite model using only preference data, namely the rewrites that are not yet separated by the existing boundary. This process is depicted by the graphical illustrations in Figure 1.

---

**Algorithm 1** Calibration Loss Calculation

---

**Require:** Threshold $t$, loss $L(\cdot)$, human text $x_h$, AI text $x_{ai}$
1: $L_h \leftarrow L(x_h)$, $L_{ai} \leftarrow L(x_{ai})$, $L \leftarrow 0$
2: $L \leftarrow L + L_h$ if $L_h < t$
3: $L \leftarrow L + L_{ai}$ if $L_{ai} > t$
4: **return** $L$

---

To determine the threshold $t$, we perform a forward pass using the rewrite model before fine-tuning on $\mathbf{X}_{train}$ and train a logistic regression model on all loss values. The threshold $t$ can be derived from the weight and the intercept of the logistic regression model. In practice, applying the calibration loss improves detection performance by 4.54% in AUROC among the 21 domains, from 0.8555 to 0.9009.

## 4 DATASET

Existing detectors are often evaluated on datasets such as SQuAD (Rajpurkar et al., 2016), XSum (Narayan et al., 2018), and Writing Prompts (Fan et al., 2018). However, these datasets typically represent a narrow subset of available data, both in terms of timeliness and domain coverage. This limitation raises concerns about over-fitting and uncertainty regarding how these detectors would perform when deployed in real-world scenarios, highlighting the necessity in creating a dataset of diversely-distributed texts for training.

### 4.1 DATA COLLECTION

To ensure the robustness and generalizability of our detection model, we construct a dataset consisting of human-written text from 21 distinct domains, including finance, entertainment, cuisine, etc. For each domain, we collect the texts either by crawling online platforms like Wikipedia or by sampling from publicly available datasets. From these collections, we randomly select 200 complete paragraphs as text snippets which yields an average length of 120 words among the samples. For each of these 200 human-written samples per domain, we generate four AI-written counterparts using four state-of-the-art LLMs: GPT-4o (OpenAI, 2024), GPT-3.5-Turbo (OpenAI, 2020), Gemini 1.5 Pro (Reid et al., 2024), and Llama-3-70B-Instruct (Meta, 2024). This results in a total of 21,000 text samples across all domains. Detailed descriptions of the domains and their sources are provided in §A.1, and examples of the dataset are shown in Figure 5.

### 4.2 PROMPT DIVERSITY

Conventionally, AI-generated text is created by prompting LLMs to either rewrite a given text or continue writing from a given prefix, often using a single, static prompt for the entire process (Mitchell et al., 2023; Bao et al., 2024; Verma et al., 2024; Mao et al., 2024). However, real-world text generation involves a wide variety of prompts, which can significantly alter the distribution of the generated text. Previous work (Mao et al., 2024) has shown that one straightforward way to bypass the RAIDAR detector is by using the prompt "Help me rephrase it, so that another GPT rewriting will cause a lot of modifications," which suggests that data generated by different prompts are different in distribution, indicating the importance of prompt diversity. To address this, we curate a dataset of 200 rewrite prompts, each containing slight variations in phrasing and instructions. For each generated text, a prompt is randomly sampled from this dataset. Examples of the prompts we use are provided below:

- Refine this for me please:
- Please rewrite this content in your own words:
- Make this text more formal and professional:

hl..

- Make this text more casual and friendly:
- Rephrase this text in a more elaborate way:
- Reframe this content in a more creative way:
- Can you make this text sound more enthusiastic?
- Rewrite this passage to emphasize the key points:
- Help me rephrase it, so that another GPT rewriting will cause a lot of modifications:

For Gemini rewrite, training on diversely-prompted dataset increases testing AUROC from 0.7302 to 0.7566. For Llama rewrite, AUROC increases from 0.7888 to 0.7970. This shows that diverse prompts effectively enables the model to better capture the distribution of AI texts in the real world, whose generation prompts are expected to vary significantly.

### 4.3 DATA CLEANING

In collecting human-written text, we ensure that no data is generated after November 30, 2022, the release date of ChatGPT (OpenAI, 2020), avoiding contamination of human dataset with AI-generated content. Instead of removing all entries that are either too short (less than 10 words) or too long (over 300 words), we retain them while maintaining an overall average length of 120 words across different domains, with standard deviation in length being 108 words. For AI-generated text, we carefully remove any extraneous suffixes, such as "Sure, here is a...," to avoid them be detected in this way.

## 5 EVALUATION

This section answers the following questions:

**Q1:** How does L2R compare with other detectors? (§5.3)
**Q2:** How does L2R perform when OOD? (§5.4)
**Q3:** How does L2R perform under adversarial attacks? (§5.5)
**Q4:** How does L2R's training objective compare with directly training for binary classification? (§5.6)
**Q5:** How does training on our proposed dataset contribute to L2R's performance? (§5.7)

### 5.1 EXPERIMENT SETUP

We perform all experiments on one NVIDIA A100 GPU with 40GB VRAM. We use 'meta-Llama/Meta-Llama-3-8B-Instruct' (AI@Meta, 2024) as the open-sourced rewrite model in all experiments. To fine-tune the Llama model with 8B parameters, we employ 4-bit QLoRA (Dettmers et al., 2024), with parameter r set to 16, lora_alpha set to 32, and lora_dropout set to 0.05, unless otherwise noted. We use an initial learning rate of 5e-6, a weight decay of 0.01, and a batch size of 32 to train until convergence. We use 70% of the dataset for training and the rest for testing in all experiments.

### 5.2 BASELINES

Our baseline detectors consist of Fast-DetectGPT (Bao et al., 2024), Ghostbusters (Verma et al., 2024), RAIDAR (Mao et al., 2024), and a custom approach named 'Llama Logits,' which involves training a Llama-3-8B model together with a classifier (same size as RAIDAR and L2R) on its logits output to perform naive text classification. For Ghostbuster, RAIDAR and 'Llama Logits', we train and test these detectors on the identical training and testing sets as L2R. For Fast-DetectGPT, we use its local version available at Fast-DetectGPT (2024). For 'Llama Logits,' we train its Llama model using the same LoRA configurations as the rewrite model in L2R for a fair comparison. We also experiment on using a close-sourced model,

Gemini 1.5 Pro (Reid et al., 2024) (referred to as Gemini Rewrite), as the rewrite model for RAIDAR in addition to Llama Rewrite. We set the sampling temperature to 0 when using Llama for rewriting during training and detection.

## 5.3 COMPARE L2R WITH OTHER DETECTORS

We compare the performance of L2R with Fast-DetectGPT, Ghostbusters, and RAIDAR (Llama Rewrite and Gemini Rewrite), by measuring the Area Under the Receiver Operating Characteristic Curve (AUROC) scores. The resulting scores for each domain along with their average and standard deviation can be found in Table 1. L2R constantly outperforms both configurations of RAIDAR in all domains; outperforms Fast-DetectGPT in 20 of 21 domains by an average of 23.04% in AUROC; and outperforms Ghostbusters in 20 of 21 domains by an average of 19.56% in AUROC. L2R has a 5.62% lower AUROC score than Fast-DetectGPT on legal document domain, and a 1.62% lower AUROC score than Ghostbusters on literature creative writing domain, which might be due to the unique distributions of these domains: legal documents require a more rigorous writing style, while creative writing has a more casual style, thus leaving fewer room for rewrite even for human writers.

In general, the fluctuating AUROC scores indicate the challenging nature of our dataset and the diversity and independence of the distributions across domains. These results also show that L2R has better knowledge of the intricate differences between human and AI texts in various domains compared with the baselines, and is more capable in the real-world setting.

## 5.4 OOD DATASET EVALUATION

We showed that L2R outperforms the state-of-the-art detectors ID in terms of AUROC scores, but it is equally important to assess its robustness under OOD conditions, as training-based detectors are prone to overfitting to familiar domains and generator models. We first evaluate this by showing its performance on OOD datasets.

To assess L2R's performance on OOD data, we adopt the M4 dataset (Wang et al., 2024), an OOD dataset that is different from our training data in multiple dimensions, including data generation models, text length, decoding strategy, and domains. We show a comparison in 2.

The results of the OOD evaluation are presented in Table 3. We include both ID and OOD results to highlight the degree of overfitting for each detector. While the Llama Logits method achieves the highest ID AUROC, its OOD result is the lowest, indicating significant overfitting to the training data. Similarly, Ghostbuster shows overfitting with its OOD AUROC being roughly half of its ID performance. The naive rewrite-based approach shows superior robustness compared with these other methods, but L2R trained with reduced parameters, i.e. rank r set to 4 and lora_alpha set to 8 (reduced params), outperforms Llama Rewrite by 3.45% ID and 4.67% OOD. This demonstrates that our fine-tuning does not simply overfits the rewrite model to the training data, but enhances its classification performance across diverse distributions.

We notice that reducing the number of training parameters make the model more generalizable, and further investigate the impact of fine-tuning parameters on L2R's performance ID and OOD. By adjusting the LoRA parameters r and lora_alpha, we define four fine-tuning configurations with the number of trainable parameters ranging from 851,968 to 6,815,744, with details listed in §A.4. Figure 3 illustrates the results, where we observe a consistent increase in ID AUROC, accompanied by a decline in OOD AUROC as the number of parameters grows. This suggests that the model becomes increasingly overfitted to the training distribution. L2R either outperforms Llama Logits OOD or both ID and OOD, and all four configurations outperform Ghostbusters and Fast-DetectGPT both ID and OOD. Also, the first two configurations surpass RAIDAR in terms of AUROC across both settings.

| Domain | Fast-DetectGPT | Ghostbusters | RAIDAR (Gemini Rewrite) | RAIDAR (Llama Rewrite) | Llama L2R |
|---|---|---|---|---|---|
| AcademicResearch | 0.4664 | 0.6597 | 0.7911 | 0.8311 | **0.8406** |
| ArtCulture | 0.6292 | 0.6781 | 0.7711 | 0.6750 | **0.8328** |
| Business | 0.6829 | 0.8331 | 0.8153 | 0.8369 | **0.9156** |
| Code | 0.6808 | 0.3770 | 0.5670 | 0.3840 | **0.8383** |
| EducationalMaterial | 0.7474 | 0.8506 | 0.9339 | **0.9675** | 0.9644 |
| Entertainment | 0.8392 | 0.8600 | 0.7836 | 0.8319 | **0.9494** |
| Environmental | 0.8382 | 0.8447 | 0.9081 | 0.9228 | **0.9786** |
| Finance | 0.6879 | 0.7828 | 0.6917 | 0.8153 | **0.9400** |
| FoodCuisine | 0.7425 | 0.6703 | 0.7181 | 0.7831 | **0.9547** |
| GovernmentPublic | 0.7100 | 0.6833 | 0.7375 | 0.7619 | **0.8675** |
| LegalDocument | **0.8365** | 0.5453 | 0.5528 | 0.6594 | 0.7803 |
| LiteratureCreativeWriting | 0.7928 | **0.9456** | 0.8056 | 0.9161 | 0.9294 |
| MedicalText | 0.5693 | 0.6242 | 0.7614 | 0.7700 | **0.7857** |
| NewsArticle | 0.5808 | 0.6800 | 0.7714 | 0.8547 | **0.9242** |
| OnlineContent | 0.6292 | 0.5922 | 0.7408 | 0.8231 | **0.8881** |
| PersonalCommunication | 0.5392 | 0.7042 | 0.6783 | 0.7233 | **0.8239** |
| ProductReview | 0.6467 | 0.7364 | 0.7150 | 0.8075 | **0.9689** |
| Religious | 0.6314 | 0.6111 | 0.7772 | 0.8397 | **0.9775** |
| Sports | 0.6015 | 0.6561 | 0.6917 | 0.7869 | **0.8742** |
| TechnicalWriting | 0.6075 | 0.7242 | 0.8269 | 0.8575 | **0.9369** |
| TravelTourism | 0.6210 | 0.7517 | 0.8492 | 0.8897 | **0.9475** |
| AVERAGE | 0.6705 | 0.7053 | 0.7566 | 0.7970 | **0.9009** |
| STD | 0.1015 | 0.1259 | 0.0928 | 0.1212 | **0.0634** |

Table 1: Comparison of detection performance measured with AUROC scores. For Ghostbuster and all rewrite-based detectors, we train a single classifier on the training set of all domains, then test the model's performance on the test set of each individual domain. **AVERAGE** measures the average performance for all independent domains, and **STD** measures the standard deviation across domains.

| Dataset | Ours | M4 |
|---|---|---|
| Generator | GPT-3.5-Turbo, GPT-4o, Llama-3-70B, Gemini 1.5 Pro | BLOOMz, ChatGPT, Davinci, Cohere, Dolly V2 |
| Text Length | Mean: 765 chars, STD: 654 chars | Mean: 1365 chars, STD: 244 chars |
| Decoding Strategy | Nucleus Sampling, Temperature = 1, top_p = 1 | Varies |
| Domains | 21 domains | 5 Non-Overlapping English domains |

**Table 2:** Comparison of characteristics of our dataset and M4 dataset, which we use for OOD evaluation.

## 5.5 ADVERSARIAL ATTACK

We employ two distinct types of attack to assess L2R's robustness against the baseline detectors. For both experiments, we apply the attack to all AI-generated texts in the testing set across all domains, while training L2R and the baselines on the unmodified training set and evaluating it on the modified testing set.

### 5.5.1 DECOHERENCE ATTACK

Bao et al. (2024) introduces the decoherence attack where two adjacent, randomly selected words are transposed in all sentences longer than 20 words within a paragraph for AI texts. Bao et al. (2024) demonstrated that this simple attack can be highly effective in degrading the performance of sate-of-the-art detectors, without affecting the core meaning of the input. We present the results of this attack in Table 4, where L2R achieves

| Model | In-Distribution | Out-of-Distribution |
|---|---|---|
| Ghostbusters | 0.7053 | 0.3888 |
| Fast-DetectGPT | 0.6705 | 0.3672 |
| Llama Logits | **0.9774** | 0.1426 |
| Llama Logits (Reduced Params) | 0.8016 | 0.3450 |
| Llama Rewrite | 0.7970 | 0.6931 |
| Llama L2R | 0.9009 | 0.6561 |
| Llama L2R (Reduced Params) | 0.8315 | **0.7398** |

Table 3: ID and OOD performance measured in AUROC scores. For L2R and Llama logits, the "Reduced Params" models are tuned with approximately 1/4 of the parameters for better generalizability. With reduced parameters, L2R has the highest OOD AUROC, outperforming the naive Llama rewrite both ID and OOD by 3.45% and 4.67%, respectively, suggesting its generalizability through fine-tuning.

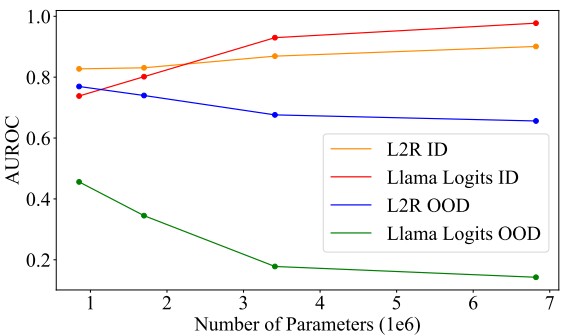

Figure 3: Relationship between the number of trainable parameters and ID and OOD AUROC scores for L2R and RAIDAR. As the number of parameters increase from $1 \times 10^6$ to $7 \times 10^6$, both L2R and RAIDAR show higher ID performance and lower OOD performance, showing how the effect of overfitting emerges as we increase the LLM's trainable parameters. L2R outperforms Llama Logits either OOD or both ID and OOD, showing the superior robustness and accuracy of L2R.

the highest AUROC on samples subjected to this attack, indicating its superior robustness compared to other models. This is because our rewrite-based objective function for fine-tuning teaches the model the innate distributions of human and AI texts, instead of relying on brittle statistical features that are easily altered through this simple attack.

### 5.5.2 REWRITE ATTACK

Mao et al. (2024) introduces the rewrite attack where a GPT-3.5-Turbo model is prompted to refine an input paragraph, generated by AI, in such a way that a subsequent rewrite by another GPT model would result in significant changes. Mao et al. (2024) showed that this type of attack is particularly effective against rewrite-based detectors, as it disrupts the rewrite we use for classification. As shown in Table 4, L2R again achieves the highest AUROC on these attack samples, further demonstrating its robustness through fine-tuning. This is because its fine-tuning objective creates separable gap between human and AI rewrite ratios that is large enough so that the attack samples remain in the AI distribution despite the perturbations. Concretely, the average edit ratio of human texts is 0.6981, and of AI texts is 0.8606. After attack, the ratio for AI decreases to 0.8386, which suggests that the rewrite attack is effective in shifting the AI distribution towards human, but there still exists a clear gap between both distributions, so that L2R's classification performance only degrades marginally.

| Model | No Attack | Decoherence Attack | Rewrite Attack |
|---|---|---|---|
| Ghostbusters | 0.7053 | 0.4730 | 0.4061 |
| Fast-DetectGPT | 0.6705 | 0.4984 | 0.5100 |
| Llama Logits | **0.9774** | 0.7281 | 0.6563 |
| Llama Rewrite | 0.7970 | 0.7681 | 0.7944 |
| Llama L2R | 0.9009 | **0.8746** | **0.8927** |

Table 4: Adversarial attack results. We employ two types of attacks, namely decoherence and adversarial rewrite. While all detectors show performance degradation under attack, L2R has the highest AUROC in both setting, suggesting its robustness through fine-tuning.

### 5.6 COMPARE L2R WITH DIRECT FINE-TUNING

A valid concern regarding L2R's superior performance is whether it is due to our fine-tuning objective, which enhances model's rewriting ability, or is solely from the fact that we exploit the vast parameters of an LLM. To answer this question, we compare L2R with the 'Llama Logits' baseline in Table 3 and 4. The Llama logits detector involves fine-tuning a Llama-3-8B model not for rewrite, but directly for binary classification.

Previously, we show that despite the Llama classifier has the highest ID AUROC score among all detectors, surpassing L2R by 7.65%, it has the lowest AUROC when evaluated OOD, up to 51.35% lower than L2R, which suggests that its performance ID is due to overfitting. This highlights the importance of our fine-tuning objective function in ensuring domain-agnostic detection accuracy. Also, the Llama classifier is inferior under adversarial attacks, with 14.65% and 23.64% lower AUROC for decoherence and rewrite attacks, respectively. This again shows L2R's robustness in capturing the true underlying distributions of human and AI data.

### 5.7 EFFECTIVENESS OF THE DIVERSE DATASET

While there exists public datasets that emphasize data diversity, including RAID (Dugan et al., 2024), RuTAD (Maloyan et al., 2022), and MAGE (Li et al., 2024), the contribution of our proposed dataset lies in its ability to train a robust and generalizable L2R model. We show this by training L2R on MAGE using the same number of texts and under the same configurations, then test its performance ID and OOD on the M4 dataset. We compare the results in 5, where L2R trained on our dataset has 15.98% higher OOD AUROC, suggesting that the diverse text distributions in our dataset is effective in training a robust and generalizable L2R model.

| Training Dataset | ID AUROC | OOD AUROC |
|---|---|---|
| MAGE | 0.8333 | 0.4963 |
| Ours | 0.9009 | 0.6561 |

**Table 5:** Comparison of L2R's ID and OOD performance when trained on MAGE and ours dataset. The superior OOD performance on our dataset suggests its effectiveness.

## 6 CONCLUSION

We present L2R, a method designed to enhance the detection of LLM-generated text by learning to rewrite more on LLM-generated inputs and less on human generated inputs. L2R excels in identifying LLM-generated content collected across various models and 21 unique domains, both ID and OOD, and under adversarial attacks. Our work demonstrates that LLMs can be trained to detect content generated by other LLMs, surpassing previous detection methods in accuracy and generalizability.

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

# A  DATASET DETAILS

## A.1  DOMAINS

Our dataset encompasses 21 indepedent domains. Below are the details for each domain in the format of domain name - source.

- AcademicResearch - Arxiv abstracts (Mao et al., 2024)
- ArtCulture - Wikipedia
- Business - Wikipedia
- Code - Code snippets (Mao et al., 2024)
- EducationalMaterial - Ghostbuster essays (Verma et al., 2024)
- Entertainment - IMDb dataset (IMDb, 2024) and Stanford SST2 (Socher et al., 2013)
- Environmental - Climate-Ins (Spokoyny et al., 2023)
- Finance - Hugging Face FIQA (Thakur et al., 2021)
- FoodCuisine - Kaggle fine food reviews (McAuley & Leskovec, 2013)
- GovernmentPublic - Wikipedia
- LegalDocument - CaseHOLD (Zheng et al., 2021)
- CreativeWriting - Writing Prompts (Fan et al., 2018)
- MedicalText - PubMedQA (Jin et al., 2019)
- NewsArticle - XSum (Narayan et al., 2018)
- OnlineContent - Hugging Face blog authorship (Schler et al., 2006)
- PersonalCommunication - Hugging Face daily dialogue (Li et al., 2017)
- ProductReview - Yelp reviews (Mao et al., 2024)
- Religious - Bible, Buddha, Koran, Meditation, and Mormon
- Sports - Olympics website (Olympics, 2024)
- TechnicalWriting - Scientific articles (Mosca et al., 2023)
- TravelTourism - Wikipedia

## A.2  GENERATION PROMPTS

Our dataset encompasses 200 different prompts for generating AI data. Here is an incomplete list of the prompts we used:

- Refine this for me please:
- Please rewrite this content in your own words:
- Make this text more formal and professional:
- Make this text more casual and friendly:
- Rephrase this text in a more elaborate way:
- Reframe this content in a more creative way:
- Can you make this text sound more enthusiastic?
- Rewrite this passage to emphasize the key points:
- Help me rephrase it, so that another GPT rewriting will cause a lot of modifications:

## A.3 EFFECTIVENESS OF THE DIVERSE PROMPT IN DATA PREPARATION

Our dataset involves 21 independent domains, four source LLMs, and 200 generation prompts, resembling real-world use cases for text detectors compared with traditional evaluation datasets which are usually constrained to one single domain and generation prompt. To prove the superiority of our dataset in training more capable detection models, we create a parallel nondiverse dataset which is created on the same number of domains and source LLMs, but generate the AI data with only with the prompt "Rewrite this for me please." Then, we train two RAIDAR detectors without fine-tuning, on the non-diverse dataset, and evaluate it on the diverse dataset. As shown in Table 6, the diverse prompts yields to 2.64% increase in AUROC score if the rewrite model is Gemini 1.5 Pro, and 0.82% increase in AUROC score if the rewrite model is Llama-3 8B. This validates the effectiveness of the diverse prompts we were using, and suggests that such diversity could help the detector to capture more information about real world data distributions.

| Dataset | Rewrite Model | AUROC |
|---|---|---|
| Single-Prompt | Gemini | 0.7302 |
| Multi-Domain Dataset | Llama | 0.7888 |
| Multi-Prompt | Gemini | **0.7566** |
| Multi-Domain Dataset | Llama | **0.7970** |

Table 6: Comparison of AUROC scores for Gemini and Llama rewrite models on nondiverse and duverse Datasets. Diverse prompting in the training set enhances detection performance for both models.

## A.4 LoRA CONFIGURATIONS FOR FINE-TUNING

Table A.4 lists the four fine-tuning configurations we use in §5.4.

| r | lora_alpha | Trainable Parameters |
|---|---|---|
| 2 | 4 | 851,968 |
| 4 | 8 | 1,703,936 |
| 8 | 16 | 3,407,872 |
| 16 | 32 | 6,815,744 |

Table 7: Parameter settings for LoRA fine-tuning.

## A.5 EFFECTIVENESS OF THE CALIBRATION LOSS

An important contribution of ours that improves the fine-tuning performance is the calibration loss, as proposed in §3.4. Without this loss, the model tends to overfit during fine-tuning as shown in Figure 4, where the model loss drastically decrease after 1500 steps, resulting in verbose rewrite even for LLM-generated text. We conduct an ablation study on five domains where the AUROC score is only 0.62 after the model overfits. We hypothesized that this technique could benefit model learning because the threshold effectively prevents further modification to model weights once an input, labeled either AI or human, falls in its respectively distribution already. Since our purpose is simply to draw a boundary rather than separate the distributions as much as possible, this halt in further weight adjustments facilitates the model to only perform parameter update on those inputs which are not yet correctly classified, so that it could converge more efficiently and effectively. Concretely, applying the calibration loss improves detection performance by 4.54% in AUROC among the 21 domains, even comparing with a model tuned with the loss before over-fitting.

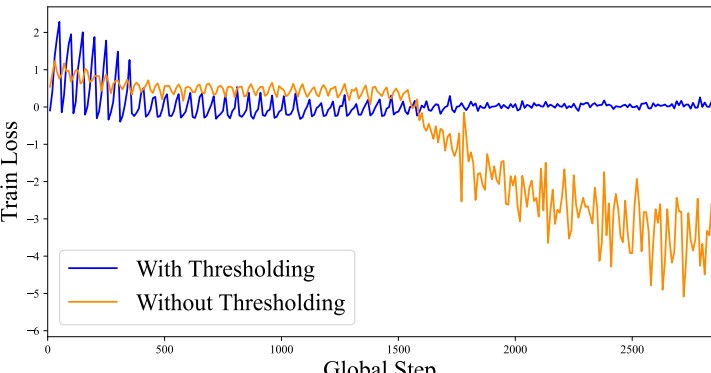

Figure 4: Training loss curves for the rewrite model. The orange plots the loss trained without the calibration method, and the blue line plots the loss trained with the method. The later one exhibits faster convergence and higher stability than the former one.

## A.6 DIFFERENT WAYS TO GENERATE OOD DATA

There exists a variety of ways to generate OOD data, including using different generation models, decoding strategies, text lengths, and writing styles. While we show how M4, the OOD dataset we use for evaluation, is distinct from our training domain in all above aspects in 2, we conduct an additional ablation study on how different text length and decoding strategy alone could influence detection performance in 8

| Avg Length | Decoding Strategy | Fast-DetectGPT | RAIDAR | L2R |
|---|---|---|---|---|
| 120 | Nucleus Sampling | 0.6833 | 0.8186 | 0.9213 |
| 60 | Nucleus Sampling | 0.6500 | 0.7635 | 0.8632 |
| 120 | Greedy Decoding & Beam Search | 0.6897 | 0.8009 | 0.8750 |

**Table 8:** Performance comparison of different setups across models.

We use 200 randomly selected texts from our dataset for both studies. For decoding strategy, we use greedy decoding for GPT and Gemini models and beam search with num_beams=5 for the Llama model. For text length, we chunk the texts to an average length of 60. We test L2R on the two datasets and show results below, where L2R outperforms RAIDAR by 9.97% for the length ablation and 7.41% for the decoding strategy ablation in AUROC. This further shows L2R's robustness to different OOD data distributions.

## A.7 EFFICIENCY OF L2R

We compare the efficiency of L2R against Llama Logits measured by FLOPS. Llama Logits only requires a forward pass of the language model, while L2R requires decoding to generate the rewrites. The FLOPS of a forward pass is 2N per token, where N is the number of parameters (Kaplan et al., 2020). Decoding is more expensive due to attention computations, but the use of KV cache has reduced this to linear complexity with respect to sequence length, thus the FLOPS per token remains approximately 2N (Pope et al., 2022). Overall, we believe we shall not sacrifice robustness just to make the algorithm fast, and our generative method is important to publish due to the improved robustness.

## A.8 REWRITE EXAMPLES

| | INPUT HUMAN
In recent years, several techniques have been proposed for increasing robustness to adversarial examples --- and yet most of these have been quickly shown to be vulnerable to future attacks. For example, over half of the defenses proposed by papers accepted at ICLR 2018 have already been broken. | REWRITTEN HUMAN
Despite recent efforts to increase robustness, most proposed techniques have been quickly shown to be vulnerable to future attacks. For instance, over half of the defenses presented at ICLR 2018 have already been breached. |
|---|---|---|
| **Academic Research** | INPUT LLAMA-3-70B
Despite numerous attempts to develop robustness against these threats, most proposed defenses have been rapidly compromised by subsequent attacks. In fact, a staggering 50% of defenses presented at ICLR 2018 have already been breached. | REWRITTEN LLAMA-3-70B
Despite numerous attempts to develop robustness against these threats, most proposed defenses have been rapidly compromised by subsequent attacks. In fact, a staggering 50% of defenses presented at ICLR 2018 have already been breached. |
| **Code**
Indentation hidden | INPUT HUMAN
def solve(s): flg = 0 idx = 0 new_str = list(s) for i in s: if i.isalpha(): new_str[idx] = i.swapcase() flg = 1 idx += 1 s = "" for i in new_str: s += i if flg == 0: return s[len(s)::-1] return s | REWRITTEN HUMAN
def solve(s): result = [] for char in s: if char.isalpha(): result.append(char.swapcase()) else: result.append(char) return "".join(result) if not any(char.isalpha() for char in s) else "".join(result)[::-1] |
| | INPUT GPT-4o
def solve(s): has_letter = any(char.isalpha() for char in s) if has_letter: return ".join(char.swapcase() if char.isalpha() else char for char in s) else: return s[::-1] | REWRITTEN GPT-4o
def solve(s): has_letter = any(char.isalpha() for char in s) if has_letter: return ".join(char.swapcase() if char.isalpha() else char for char in s) else: return s[::-1] |
| **Legal Document** | INPUT HUMAN
This type of information may constitute trade secrets. See G.L.c. 266, §30 (defining "trade secret" as used in G.L.c. 93, §42 | REWRITTEN HUMAN
This type of information may constitute trade secrets. Massachusetts General Laws, chapter 266, section 30 |
| | INPUT GEMINI 1.5 PRO
This type of information, such as customer data and proposals, can be legally protected as trade secrets under Massachusetts law (G.L.c. 266, §30). | REWRITTEN GEMINI 1.5 PRO
This type of information, such as customer data and proposals, can be legally protected as trade secrets under Massachusetts law (G.L. c. 266, § 30). |
| **Creative Writing** | INPUT HUMAN
A voice in the sky will tell you when you've left a place for the last time. You tell your SO goodbye and head out to [ location of your choice ]. Halfway there you hear "You have visited a location for the last time." John wrinkled his nose and cleared his throat. Pausing, he put down his briefcase and adjusted his tie. It was a red tie with white stripes. This was one of his favorite ties. | REWRITTEN HUMAN
As John bid his SO farewell, he headed out to [location of his choice]. Halfway there, a voice in his mind whispered, "You have visited this place for the last time." John's expression faltered, and he paused to adjust his tie, a favorite red tie with white stripes. |
| | INPUT LLAMA-3-70B
As John bid farewell to his partner and headed out to the city, a mysterious voice in the sky announced, "You have visited a location for the last time." He paused, adjusting his favorite red tie with white stripes, and cleared his throat. | REWRITTEN LLAMA-3-70B
As John bid farewell to his partner and stepped out into the city, a mysterious voice in the sky announced, "You have visited a location for the last time." He paused, adjusting his favorite red tie with white stripes, and cleared his throat. |

Figure 5: Examples of texts in our proposed dataset along with the amount of edits L2R model gives for human and LLM data. Deleted characters are marked in red, inserted characters are in blue, and unmodified characters are in **black**. The examples demonstrate the diverse domains and source LLMs available in the dataset, as well as L2R's ability in separating human and LLM texts via rewriting.

