# OpenReview forum: "Learning to Rewrite: Generalized Detection of LLM-Generated Text"
_ICLR.cc/2025/Conference — Submitted to ICLR 2025_

### Official Review · Reviewer_DZDQ · 2024-11-03

**Soundness:** 2
**Presentation:** 1
**Contribution:** 2
**Rating:** 5
**Confidence:** 5

**Summary:**

The paper introduces a novel method for detection of AI-generated text. The method involves finetuning a LLaMA 8B model to rewrite its input text such that human-written text gets rewritten quite a lot and AI-generated text gets re-written very little. At inference time, by thesholding the normalized Levenshtein distance between the input text sequence and the sequence outputted by the finetuned LLaMA, a prediction is made of either "AI-generated" or "human-written." In experiments, this method outperforms baseline approaches.

**Strengths:**

The problem of detection of AI-generated text is very important, and the proposed method tackling it is not one I had seen before. The method is described in sufficient detail I could probably reproduce the main ideas.

**Weaknesses:**

## Weaknesses of the Proposed Method
1. I am concerned with how expensive the proposed method is. Doing hundreds of tokens of generation using an 8B model in order to create a single binary prediction feels extremely inefficient. "Llama Logits" is a much more efficient approach since it only needs to do a single prediction. I would like to see a figure plotting the avg number of FLOPs each method uses for a single prediction against the method's performance at the task.
2. This is more of a question than an obvious weakness, but why choose the method described in 3.2 for regularization instead of just adding a second term in the loss which is the standard next-token prediction language modeling loss (similar to what RLHF does to keep the RLHF'ed model from straying from the reference model)?

## Weaknesses of the Experimental Design
1. There are key flaws in the baselines L2R is compared against. Most concerning is line 260: "For ’Llama Logits,’ we train its Llama model using the same LoRA configurations as the rewrite model in L2R for a fair comparison." If I understand this correctly, the authors did hyperparamter search to find a good finetuning configuration configuration for *their* method (L2R), and then applied this same configuration to the Llama Logits baseline. This is the opposite of a fair comparison; a fair comparison would be to use equal compute and effort to identify a good hparam configuration for the Llama Logits baseline as for L2R. From the third line in Table 2, it is apparent that the authors did not find a good set of hyperparameters for tuning LLama Logits, as the discrepancy between between in-distribution and out-of-distribution performance suggests considerable overfitting.
2. The paper does not explain what decoding strategy was used for generating the LLM-generated examples. This can make a big difference in terms of detectability, so it is very important to report this. I would expect this to be mentioned in Section 4.1.
3. It is also unclear what prompts were used for generated the LLM-generated examples. This should also be mentioned in Section 4.1.
4. I would like to see some text added to motivate why an average sequence length of 120 words was chosen for the task.
5. In Appendix A.3, the authors say "to prove the superiority of our dataset in training more capable detection models, we create a parallel nondiverse dataset ..." This statement doesn't sit well with me since the authors are only proving the superiority of their dataset over an obviously worse version of their dataset. A more valid comparison would be to compare their dataset to other publicly available datasets intended for the detection task.
6. There are many ways for eval data to be OOD for a detection system: it could come from a different model than the one used to collect training data for the detection system; it could be shorter or longer than the training data; it could be generated using a different decoding strategy; it could be in a different writing style (e.g. news vs. stories). The authors only focus on this very last definition of OOD; I would like to see at least some exploration or discussion of other ways eval data could be OOD.

## Weaknesses of the Discussion of Prior Work
1. It is unclear from the Related Work section how the proposed method differs from prior methods, especially RAIDAR, which has a very similar core idea. I would like the Related Work section to contain more sentences along the line of "In contrast to Paper X which does approach Y, we do approach Z." For example, I cannot parse what is meant by the one sentence in the related work section that does attempt to compare to RAIDAR (line 130): "Despite the attempt on capturing rewrite edit distance as a domain-agnostic feature, the rewrite amount still varies across distributions, which limits its full potential." Why does the rewrite amount differing across distributions limit RAIDAR? What do the authors do differently in their approach to solve this?
2. The only pre-2020 paper referencd in the Related Work section is to the GPT-2 open-weight release whitepaper, despite there being several seminal works on detection of generated text which came out around then. A couple notable omissions are GLTR (https://arxiv.org/abs/1906.04043) and "Automatic Detection is Easiest when Humans are Fooled" (https://arxiv.org/abs/1911.00650), although there are undoubtedly otfhers. The authors need to expand their literature review (and possibly their choice of baselines), as several of these simple methods from the (slightly) older literature continue to work quite well under some conditions, as can be seen in the RAID paper (https://arxiv.org/abs/2405.07940), which is another missing citation.
3. The Datasets section starts with the following sentences: "Existing detectors are often evaluated on datasets such as SQuAD (Rajpurkar et al., 2016), XSum (Narayan et al., 2018), Writing Prompts (Fan et al., 2018), and others (Bao et al., 2024; Mao et al., 2024). However, these datasets typically represent a narrow subset of available data, both in terms of timeliness and domain coverage." There are two issues with these sentences. First, "datasets ... such as others" is not an informative statement. If the authors plan to site Bao and Mao, they should list what type of data these methods evaluated on. Second, this list is missing several key detection benchmarks, which are also trying to solve the "real-world scenarios" challenge that the authors mentioned being concerned about in the next sentence in the paragraph. Missing references include RAID (linked above), RuATD (https://arxiv.org/abs/2206.08029), MAGE (https://arxiv.org/abs/2305.13242), and probably others as well. The authors should explain what their new benchmark accomplishes that these existing benchmarks do not.

## Weaknesses in the Writing/Presentation
1. Equations 3 could be greatly simplified by formulating the problem such that label y = 1 is AI and **y = -1 is (human)**. This would eliminate the need for the indicator function and the funky arithmetic.
2. Equation 4 reads like a Python program which the authors attempted to turn into math. It would be much more comprehensible if the authors instead wrote what this in pseudocode, or even just English.
3. In Table 1, under "EducationalMaterial" the bolded value should be "Llama Rewrite" not "Llama L2R." Speaking of which, is the difference between the values here statistically significant?
4. Many of the references are formatted incorrectly with missing capitalization in the paper titles.

**Questions:**

See above.

---

> ### Author Response · Authors · 2024-11-25
>
> Thank you for the thoughtful review. We are glad that the reviewer found the problem important and our method innovative.
>
> ### On Proposed Method
> 1. **Efficiency of L2R Versus Llama Logits** Thank you for the suggestion. While Llama Logits is more efficient, it significantly overfits when OOD (Table 2), whereas L2R maintains consistent performance ID and OOD. The FLOPS of a forward pass is 2N per token, where N is the number of parameters [1]. Decoding is more expensive due to attention computations, but the use of KV cache has reduced this to linear complexity with respect to sequence length [2], thus the FLOPS per token remains approximately 2N. Overall, we believe we shall not sacrifice robustness just to make the algorithm fast, and our generative method is important to publish due to the improved robustness. We will add the efficiency comparison in revision.
>
> 2. **Analogy with RLHF** Thank you for bringing the angle of RLHF into discussion. We would like to note that the calibration loss employs a similar idea to DPO in RLHF, where we fine-tune the rewrite model using only preference data, namely the rewrites that are not yet separated by the existing boundary. We will add this to the discussion.
>
> ### On Experimental Design
> 1. **Fairness of Evaluation** We respectfully disagree with the reviewer’s concern on fairness. On line 3 and 6 of Table 2, the comparison between Llama Logits and L2R was based on the same set of LoRA parameters, which we used the maximum combination without OOM. Still, we followed the reviewer’s suggestion and did a hyperparameter search, and show results below.  L2R consistently outperforms Llama Logits either under OOD, or both ID and OOD, confirming its robustness across settings. We will add this in revision.
>
> | lora_r | lora_alpha | Parameters | Llama Logits ID | Llama Logits OOD | L2R ID | L2R OOD |
> |--------|------------|------------|-----------------|------------------|--------|---------|
> | 2      | 4          | 8.52E+05   | 0.7381          | 0.4559           | 0.8273 | 0.7694  |
> | 4      | 8          | 1.70E+06   | 0.8016          | 0.3450           | 0.8315 | 0.7398  |
> | 8      | 16         | 3.41E+06   | 0.9298          | 0.1780           | 0.8691 | 0.6761  |
> | 16     | 32         | 6.82E+06   | 0.9774          | 0.1426           | 0.9009 | 0.6561  |
>
> 2. **Impact of Different Decoding Strategies** Thank you for the question on decoding strategy. To generate the dataset, we used the default strategy, namely nucleus sampling with temperature and top_p set to 1 for random generation, which is the most widely used. To rewrite, we use greedy decoding for deterministic outputs. Following the reviewer’s suggestion, we did an ablation study on 200 random texts from our dataset generated using distinct decoding strategies for generation. We use greedy decoding for GPT and Gemini models and beam search with num_beams=5 for the Llama model. We show results below, where L2R outperforms RAIDAR by 7.41% in AUROC. We will add this in revision.
> | Setup                            | Fast-DetectGPT | RAIDAR  | L2R    |
> |----------------------------------|----------------|---------|--------|
> | Nucleus Sampling | 0.6833         | 0.8186  | 0.9213 |
> | Greedy Decoding & Beam Search  | 0.6897         | 0.8009  | 0.8750 |
>
> 3. **Examples of generation prompts** We have a list of prompts in Appendix A.2, and we will move it to the main paper.
>
> 4. **Question on Average Text Length** Sorry for the confusion. When crawling online, we segment longer text into complete paragraphs without further chunking, and the text lengths in our dataset vary significantly across domains with an average of 120 and standard deviation of 108. We will clarify this in revision.
>
> 5. **Purpose of the Ablation Study of Diverse Prompting** Thank you for bringing this up. The goal of this ablation study was to measure the impact of diverse prompting on detection. Non diverse prompting was the prompting method used to generate testing data for multiple previous detectors, including RAIDAR, Fast-DetectGPT, and Ghostbusters.
> **Comparison with Public Dataset** To demonstrate our proposed dataset’s superiority over the open source datasets in training L2R, we train L2R on MAGE [3] using the same configurations, and test ID and OOD on the M4 dataset. When trained on our dataset, L2R exhibits 15.98% higher OOD AUROC, which proves our dataset’s effectiveness in training a more generalizable detection model. We will add this comparison in revision.
> | Training Dataset | ID     | OOD    |
> |-------------------|--------|--------|
> | MAGE              | 0.8333 | 0.4963 |
> | Ours              | 0.9009 | 0.6561 |

---

> > ### Author Response · Authors · 2024-11-25
> >
> > 6. **Different ways to create OOD data** Thank you for the suggestion. The M4 dataset that we use in the OOD evaluation already satisfies all the requirements from the reviewer, as we show below.
> > | Dataset              | Ours                                                                 | M4                                 |
> > |--------------------------|-----------------------------------------------------------------------------|-----------------------------------------------|
> > | Data Generation Models | GPT-3.5-Turbo, GPT-4o, Llama-3-70B, and Gemini 1.5 Pro                      | BLOOMz, ChatGPT, Davinci, Cohere, and Dolly V2 |
> > | Text Length           | Mean: 765 chars, STD: 654 chars                                             | Mean: 1365 chars, STD: 244 chars              |
> > | Decoding Strategy     | Nucleus Sampling, Temperature = 1, top_p = 1                                | Varies                                        |
> > | Domains               | 21 domains                                                                 | 5 Non-overlapping English domains             |
> >
> > Still, following the suggestion of the reviewer, we did additional ablation studies on decoding strategy and text length. We use 200 randomly selected texts from our dataset for both studies. For decoding strategy, we use greedy decoding for GPT and Gemini models and beam search with num_beams=5 for the Llama model. For text length, we chunk the texts to an average length of 60. We test L2R on the two datasets and show results below, where L2R outperforms RAIDAR by 9.97% for the length ablation and 7.41% for the decoding strategy ablation in AUROC. We will add this in revision.
> > | Setup                            | Fast-DetectGPT | RAIDAR  | L2R    |
> > |----------------------------------|----------------|---------|--------|
> > | Avg Length = 120, Nucleus Sampling | 0.6833         | 0.8186  | 0.9213 |
> > | Avg Length = 60, Nucleus Sampling  | 0.6500         | 0.7635  | 0.8632 |
> > | Avg Length = 120, Greedy Decoding & Beam Search  | 0.6897         | 0.8009  | 0.8750 |
> >
> > ### On Discussion of Prior Work
> > 1. **Contributions of L2R** Sorry for the confusion. RAIDAR was not trained and tested on diverse domains, whereas L2R outperforms it in our diverse evaluation. Additionally, L2R’s fine-tuning objective ensures an universal edit ratio threshold across domains and requires only one classifier, making it more generalizable than RAIDAR, which produces varying edit ratio thresholds across domains and requires separate classifiers. We will make this more clear in revision.
> >
> > 2. **Missing Citations in Related Works** Thank you for the suggestions, we will add the citations.
> >
> > 3. **Writing and Missing Citations** We will update the statement and add the citations. Please refer to Experimental Design point 5 for the contribution of our dataset.
> >
> > ### On Writing/Presentation
> > Thank you for the suggestions, we will update the equations, Table 1, and citations. The difference in average AUROC between Llama Rewrite and Llama L2R is statistically significant, with a p-value of 6.77e-141.
> >
> > [1] Kaplan, J., McCandlish, S., Henighan, T., Brown, T. B., Chess, B., Child, R., Gray, S., Radford, A., Wu, J., & Amodei, D. (2020). Scaling Laws for Neural Language Models. arXiv. https://arxiv.org/abs/2001.08361
> >
> > [2] Pope, R., Douglas, S., Chowdhery, A., Devlin, J., Bradbury, J., Levskaya, A., Heek, J., Xiao, K., Agrawal, S., & Dean, J. (2022). Efficiently Scaling Transformer Inference. arXiv. https://arxiv.org/abs/2211.05102
> >
> > [3] Li, Y., Li, Q., Cui, L., Bi, W., Wang, Z., Wang, L., Yang, L., Shi, S., & Zhang, Y. (2024). MAGE: Machine-generated Text Detection in the Wild. ACL. https://arxiv.org/abs/2305.13242

---

> ### Comment · Reviewer_DZDQ · 2024-11-26
> **I appreciate the authors' efforts to address my concerns about the paper.**
>
> However, given the number of new results and other revisions they are proposing, I think this paper would benefit from another round of review.

---

> ### Author Response · Authors · 2024-11-26
>
> Dear reviewer,
> The tables we showed in the rebuttal are not to introduce new results but to support our clarification texts regarding your concerns. We truly appreciate your detailed feedback from the review cycle, and we are confident that the texts in our rebuttal address your concerns. If the reviewer agrees that all the concerns have been addressed, we believe our paper should be ready to publish. If the reviewer has any further questions that are left unanswered, please let us know.

---

### Official Review · Reviewer_XFFn · 2024-11-03

**Soundness:** 3
**Presentation:** 3
**Contribution:** 3
**Rating:** 6
**Confidence:** 4

**Summary:**

Learning2Rewrite (L2R) is an innovative framework for detecting AI-generated text by exploiting LLMs' tendency to modify human-written content more extensively than AI-generated text during rewriting. The framework's core innovation lies in its training objective, which minimizes edits on AI-generated text while maximizing changes to human-written content, creating a clear classification boundary. A calibration loss mechanism prevents overfitting and ensures stable performance across domains.

Comprehensive evaluations across 21 domains using GPT-3.5, GPT-4, Gemini, and Llama-3 demonstrate L2R's effectiveness. It surpasses existing detectors like RAIDAR and Fast-DetectGPT with up to 23.04% higher AUROC in in-distribution tests and 37.26% in out-of-distribution scenarios. The framework is robust against adversarial attacks and effective generalization to new domains, validated through a diverse evaluation dataset that serves as a reliable benchmark for AI text detection methods.

**Strengths:**

**Originality**: L2R introduces two key innovations in AI text detection: using LLMs' rewriting behaviour as a detection mechanism rather than traditional classification and implementing a training objective that minimizes AI text edits while maximizing human text edits. Enhanced by a calibration loss mechanism, this approach offers a fundamentally new way to distinguish between human and AI-generated content.

**Quality**: The evaluation spans 21 domains using GPT-3.5, GPT-4, Gemini, and Llama-3, with L2R outperforming RAIDAR and Fast-DetectGPT on both in-distribution and out-of-distribution tasks. The method is robust against adversarial attacks, and its effectiveness is validated through comprehensive ablation studies examining parameter impacts and training configurations.

**Clarity**: The paper presents its technical contributions precisely and clearly. The methodology and training objectives are thoroughly documented and supported by illustrative visualizations of edit distance distributions. The experimental setup and results are systematically organized, providing clear evidence for the method's performance.

**Significance**: L2R advances AI text detection through improved cross-domain generalization and adversarial robustness. Its interpretable detection mechanism and practical effectiveness in identifying AI-generated content make it particularly valuable for real-world applications in misinformation detection.

Overall, I like this paper's approach, which presents an elegant and effective solution for AI text detection.

**Weaknesses:**

- The paper's robustness evaluation, while covering decoherence and rewriting attacks, could benefit from exploring more adversarial scenarios. Testing against AI text modified by advanced paraphrasing tools or examining mixed human-AI content would provide deeper insights into L2R's limitations.

- Moreover, while L2R's success relies on diverse training data and prompt variations, the paper would benefit from an analysis of how reduced data diversity affects its performance.

- The method appears limited in handling cases involving mixed human and AI-authored text, where the task is to identify specific AI-generated segments. This limitation could be significant, as human-AI collaborative writing is increasingly common. Addressing this challenge would broaden the method's applicability and practical value.

**Questions:**

How does the calibration loss mechanism specifically prevent overfitting compared to standard training? While Figure 4 in the appendix illustrates training loss behavior, how does this confirm the effectiveness of the calibration loss? Shouldn't the impact of preventing overfitting be more evident in the experiments on out-of-distribution test set performance, with or without the use of calibration loss?

---

> ### Author Response · Authors · 2024-11-25
>
> Thank you for the thoughtful review. We are glad that the reviewer found our method innovative, our experiment comprehensive, and our writing precise and clear.
>
> ### On Weaknesses
> 1. **Adversarial Attacks** Thank you for bringing this up. We choose two realistic adversarial attacks that produce high-quality texts, which align with the baselines [1] and [2]. We show the results below, where L2R outperforms the baselines under decoherence and rewrite attack by up to 37.62% and 38.27% respectively.
> | Model             | No Attack | Decoherence Attack | Rewrite Attack |
> |-------------------|-----------|--------------------|----------------|
> | Fast-DetectGPT    | 0.6705    | 0.4984             | 0.5100         |
> | Llama Rewrite     | 0.7970    | 0.7681             | 0.7944         |
> | Llama L2R         | 0.9009    | 0.8746             | 0.8927         |
>
> 2. **Ablation study on data diversity** Thank you for the suggestion. We have an ablation study on data diversity in Appendix A.3. We control data diversity by different number of generation prompts, and training on data generated by 200 prompts increases AUROC by 2.64% and 0.82% for Gemini and Llama rewrite, respectively. We will move the table to the main paper.
> | Dataset                           | Rewrite Model | AUROC   |
> |-----------------------------------|---------------|---------|
> | Single-Prompt | Gemini        | 0.7302  |
> | Single-Prompt | Llama         | 0.7888  |
> | Multi-Prompt | Gemini        | 0.7566  |
> | Multi-Prompt | Llama         | 0.7970  |
>
> 3. **Detection of Human-AI Collaborative Writing** Indeed it is quite challenging to detect texts written by human-AI collaboration, but our framework could be applied to the task by fine-tuning on collaborative AI data and analyzing hybrid patterns in edit distances for text segments. Human-AI collaboration is out of the scope of this paper.
>
> ### On Questions
> 1. **Mechanism of the Calibration Loss** Sorry for the confusion. Concretely, the calibration loss enables the model to only optimize against the hard examples, and leave those already correctly classified unchanged, so that we prevent overfitting. We will make this more clear in revision.
>
> 2 & 3. **Ablation study on Calibration Loss** Sorry for the confusion. We showed ablation study results in section 3.3, where fine-tuning with calibration loss yields to 4.54% higher AUROC. We format the results in the table below, and will move it to the main paper.
> | Model       | With Calibration Loss | Without Calibration Loss |
> |--------------|------------------------|---------------------------|
> | AUROC    | 0.9009                 | 0.8555                    |
>
> [1] Bao, G., Zhao, Y., Teng, Z., Yang, L., & Zhang, Y. (2024). Fast-DetectGPT: Efficient Zero-Shot Detection of Machine-Generated Text via Conditional Probability Curvature. ICLR. https://arxiv.org/abs/2310.05130
>
> [2] Mao, C., Vondrick, C., Wang, H., & Yang, J. (2024). Raidar: geneRative AI Detection viA Rewriting. ICLR. https://arxiv.org/abs/2401.12970

---

### Official Review · Reviewer_8px3 · 2024-11-04

**Soundness:** 2
**Presentation:** 3
**Contribution:** 2
**Rating:** 5
**Confidence:** 4

**Summary:**

This paper proposes the L2R framework for detecting AI-generated text, leveraging the insight that LLMs inherently modify AI-generated content less than human-written text when tasked with rewriting. By fine-tuning an LLM to amplify this tendency, the L2R framework demonstrates significantly improved performance across diverse domains and even under adversarial conditions.

**Strengths:**

- The paper is clear and easy to follow.
- The approach of using LLM edit distance to detect AI-generated text is innovative.
- The experiments demonstrate the method’s strong performance and robustness to some extent.

**Weaknesses:**

(1) The motivation behind the dataset collection process is unclear. Although the proposed dataset spans various domains, it is flawed because the AI-generated text is initially written by humans and then revised by an LLM—similar to a non-native speaker using AI to polish their writing. While detecting AI-generated content is important, I believe using LLMs specifically to rewrite text is one of the less risky applications. Thus, I’m not convinced that this dataset adds substantial value for benchmarking models that detect LLM-generated content.

(2) The superior performance of Llama logits on ID setting, and its poor performance on OOD setting confirms that there’s a gap between the dataset built by the authors and the real-world scenario of LLM-generated content detection. (The ID performance of Llama logits is also skipped in Table 1).

(3) I would recommend the authors to compare L2R with baseline methods on conventional datasets such as XSum, SQuAD, and WritingPrompts; due to the dataset proposed in the paper only contains LLM rewritten text.

**Questions:**

(4) Would you explain why the Fast-DetectGPT performance drops so much on OOD examples, since it doesn’t require any training?

(5) In DetectGPT paper, their hypothesis is
> Minor rewrites of model-generated text tend to have lower log probability un- der the model than the original sample, while minor rewrites of human-written text may have higher or lower log prob- ability than the original sample.

Do your findings agree or contradict with their hypothesis?

---

> ### Author Response · Authors · 2024-11-25
>
> Thank you for the thoughtful review. We are glad that the reviewer found our paper easy to follow and our method innovative, with strong performance and robustness.
>
> 1. **Motivation for Dataset Collection** Thank you for the suggestion. We acknowledge that having different ways to generate AI datasets would benefit data diversity. However, as we show in the OOD evaluation, L2R trained on our dataset has good generalizability on the M4 dataset, where AI data was not generated using rewrite but in a question answering fashion [1]. This suggests that our dataset is capable of training a generalizable detection model, which proves the value of our proposed dataset.
>
> 2. **Gap with Real-World Texts** Thank you for bringing this up. Our paper shows by training on our in-house data, our detection model can generalize to the real world. Instead of stating that our proposed dataset is an universal representation of all human and AI data, we deliberately set up the OOD evaluation on a different distribution to test the generalizability of our model, and we achieved 4.67% higher AUROC compared with RAIDAR. We will make this more clear in revision.
>
> **Comment: Contribution of Our Dataset** To further demonstrate our proposed dataset’s value in training a robust detection model, we train L2R on MAGE [2] using the same configurations, and test ID and OOD on the M4 dataset. When trained on our dataset, L2R exhibits 15.98% higher OOD AUROC, which proves our dataset’s effectiveness in training a more generalizable detection model. We will add this comparison in revision.
> | Training Dataset | ID     | OOD    |
> |-------------------|--------|--------|
> | MAGE              | 0.8333 | 0.4963 |
> | Ours              | 0.9009 | 0.6561 |
>
> 3. **Comparison on Conventional Datasets** We have included XSum and Writing Prompts datasets, as listed in Appendix A.1, and we show the results below. Additionally, our OOD evaluation proves the strong performance of L2R on non-rewritten texts in the M4 dataset.
> | Domain           | Fast-DetectGPT | Ghostbusters | Llama Rewrite | Llama L2R |
> |-------------------|----------------|--------------|---------------|-----------|
> | XSum      | 0.5808         | 0.6800       | 0.8547        | 0.9242    |
> | Writing Prompts  | 0.7928         | 0.9456       | 0.9161        | 0.9294    |
>
> 4. **Fast-DetectGPT OOD Performance** Thank you for bringing this up. There are two reasons. First, Fast-DetectGPT relies on the logits of a GPT-2 model and employs reference scores for probability estimation, which mainly come from the GPT family, while AI data in M4 are generated by models from other families, including Dolly, Cohere, BLOOMZ, and FLAN-T5. Second, the resulting AI scores from Fast-DetectGPT varies across domains, and its inability to find a universal threshold leads to the low AUROC, which is exactly the problem L2R is trying to solve. To confirm this, we show Fast-DetectGPT’s strong AUROC on subsets that are solely generated by ChatGPT, indicated by Ours (ChatGPT) and M4 (ChatGPT).
> | Dataset             | Ours (ChatGPT) | Ours (All) | M4 (ChatGPT) | M4 (All) |
> |---------------------|----------------|------------|--------------|----------|
> | Fast-DetectGPT AUROC| 0.7636         | 0.6705     | 0.9933       | 0.3672   |
>
> 5. **Agreement with DetectGPT Hypothesis** Our findings align with DetectGPT’s hypothesis. A text having greater edit distance from its LLM rewrite would have lower log likelihood, and vice versa. This motivated us for fine-tuning the rewrite model to find a consistent threshold for rewrite distance across different distributions.
>
> [1] Wang, Y., Mansurov, J., Ivanov, P., Su, J., Shelmanov, A., Tsvigun, A., Whitehouse, C., Afzal, O. M., Mahmoud, T., Sasaki, T., Arnold, T., Aji, A. F., Habash, N., Gurevych, I., & Nakov, P. (2024). M4: Multi-generator, Multi-domain, and Multi-lingual Black-Box Machine-Generated Text Detection. EACL. https://arxiv.org/abs/2305.14902
>
> [2] Li, Y., Li, Q., Cui, L., Bi, W., Wang, Z., Wang, L., Yang, L., Shi, S., & Zhang, Y. (2024). MAGE: Machine-generated Text Detection in the Wild. ACL. https://arxiv.org/abs/2305.13242

---

> > ### Comment · Reviewer_8px3 · 2024-11-26
> > **I updated my score.**
> >
> > Thank you for your response. The additional results and explanation addresses some of my concerns. Therefore I increased my score. However, beside the numbers being promising, it is still unclear why and how fine-tuning on the proposed data set can improve the OOD performance. And whether it has any side effects.

---

> > > ### Author Response · Authors · 2024-11-26
> > > **Good question which addresses why L2R is neat**
> > >
> > > That’s a really good question whose answer addresses why L2R is a neat idea - 1. Fine-tuning on rewriting generation, which makes deltas between Human and LLM texts more distinguishable against each other so that our detector learns to achieve an invariant threshold across several domains of data 2. Fine-tuning on a diversely generated dataset, which elaborates the deltas among distinct domains (e.g., legal vs. creative writing) to the model and thus boosts generalization on more unseen domains that might have a similar range of deltas. Really appreciate the reviewer for the question!

---

### Official Review · Reviewer_NyWU · 2024-11-10

**Soundness:** 3
**Presentation:** 2
**Contribution:** 2
**Rating:** 6
**Confidence:** 3

**Summary:**

This work proposes a novel method, coined Learning2Rewrite, which employs an LLM to rewrite an input text and determines whether it is written by human or a generative model based on the differences between the original text and the rewrite. Instead of training a classifier, it enhances an LLM in such a way that it makes few edits if a text is written by a model, while makes substantial edits if a text is written by human. Their experiments on data from 21 domains demonstrate the effectiveness of this approach.

**Strengths:**

* The proposed method is simple and effective.
* The experiments mitigate potential domain bias by covering the data in 21 domains.
* Most parts of the manuscript are easy-to-follow.
* There are nice illustrations, such as Figure 1, which help understand the key ideas.
* The examples in Figure 2 are great qualitative examples for understanding the effectiveness of the method.
* It is great to consider dfferences between prompts in the experiments.

**Weaknesses:**

* It lacks of an ablation study to justify the effectiveness of the calibration loss.
* It is unclear to me to what extend the fine-tuning is useful. I cannot find the details of Llama Rewrite so that it is not clear how well the fine-tuned model is compared with the one without fine tuning.

**Questions:**

* Why edit distance is choosen for similarity comparision? Why not use the other similarity measures, e.g. MAUVE?
* Is the proposed method model agnostic? It would be great to learn if the proposed method is applicable to the other model families.
* Table 2 shows that Llama L2R performs better than Llama Rewrite only with reduced parameters in the OOD setting. How does the OOD performance vary across domains?

---

> ### Author Response · Authors · 2024-11-25
>
> Thank you for the thoughtful review. We are glad that the reviewer found our method simple and effective, our experiment comprehensive, and our paper easy to follow.
>
> ### On Weaknesses
> 1. **Ablation Study on Calibration Loss** Thank you for the question. We showed ablation study results in section 3.3 to demonstrate the effectiveness of our calibration loss. L2R trained with calibration loss has 4.54% higher AUROC. We format the results in the table below, and will move it to the main paper.
> | Model       | With Calibration Loss | Without Calibration Loss |
> |--------------|------------------------|---------------------------|
> | AUROC    | 0.9009                 | 0.8555                    |
>
> 2. **Usefulness of Fine-Tuning** Sorry for the confusion. Llama Rewrite is L2R without fine-tuning the rewrite model. We summarize the results with and without fine-tuning below, where fine-tuning enhances AUROC by 3.45% ID and 4.67% OOD. We will make this more clear in revision.
> | Model           | ID     | OOD    |
> |------------------|--------|--------|
> | w/o fine-tuning | 0.7970 | 0.6931 |
> | w/ fine-tuning  | 0.8315 | 0.7398 |
>
> ### On Questions
> 1. **MAUVE as Similarity Measure** Thank you for the suggestion. We choose edit distance over MAUVE because edit distance is for individual samples, while MAUVE is for measuring distribution similarity [1]. Still, following the suggestion by the reviewer, we show the MAUVE results below, and observe a noticeable distinction between human and AI.
> | MAUVE | Human  | AI     |
> |-------|--------|--------|
> | ID    | 0.2688 | 0.6012 |
> | OOD   | 0.0041 | 0.9958 |
>
> 2. **Model Agnosticity** Our method is model agnostic. We modified the training objective, not the architecture, therefore it works with all existing LLM architectures. In addition, our OOD experiments shows that training on data generated by GPT-3.5-Turbo, GPT-4o, Llama-3-70B, and Gemini 1.5 Pro will enable detecting text generated from BLOOMz, ChatGPT, Davinci, Cohere, and Dolly V2, showing detection is model agnostic to the model that generate the text.
>
> 3. **OOD Performance Across Domains** Thank you for bringing this up. Our result in table2 shows that L2R has superior performance than Llama rewrite OOD. We show OOD performance across domains below, where Llama L2R overperforms Llama Rewrite on 4 of the 5 domains.
> | Domain          | Llama Rewrite | Llama L2R |
> |------------------|---------------|-----------|
> | Wikipedia       | 0.7375        | 0.7225    |
> | Reddit ELI5     | 0.7075        | 0.7100    |
> | WikiHow         | 0.6925        | 0.7200    |
> | PeerRead        | 0.5425        | 0.6750    |
> | Arxiv Abstract  | 0.8325        | 0.8825    |
>
> [1] Pillutla, K., Swayamdipta, S., Zellers, R., Thickstun, J., Welleck, S., Choi, Y., & Harchaoui, Z. (2021). MAUVE: Measuring the Gap Between Neural Text and Human Text using Divergence Frontiers. NeurIPS. https://arxiv.org/pdf/2102.01454

---

### Author Response · Authors · 2024-11-25
**We made clarifications, ran support experiments asked by the reviewer, and updated the paper. Edits are in blue.**

We sincerely thank all reviewers for their thoughtful and insightful feedback. We are encouraged by the recognition of the effectiveness of L2R yet its being easy to follow and the contribution of our diversely created dataset. We made clarifications, ran support experiments asked by the reviewer, and integrated them to strengthen our paper. We indicate our paper edits in blue.

---

### Comment · Area_Chair_h56N · 2024-11-27

Dear reviewers,

Thank you for your efforts reviewing this paper. If you haven't, can you please check the authors' responses and see if your concerns have been addressed? Please acknowledge you have read their responses. Thank you!

---

### Author Response · Authors · 2024-12-03
**Summarization of L2R's contributions and the rebuttal**

*Our Main Contributions:*

1. Fine-tuned detectors for generated text detection are known for overfitting to specific domains. We propose L2R, whose learning objective is rather to enlarge the edit distance between rewriting and the original text for LLM-generated text while minimizing the ones that are human-generated. This learning objective is relatively domain-agnostic, yielding an invariant detection threshold across different data distributions.

2. We build a diversely generated dataset (21 domains) and design a calibration loss function to make fine-tuning both effective and stable.

3. We conduct comprehensive evaluations on ID, OOD datasets and against different adversarial attacks (Decoherence and Rewrite bypassing), showing that L2R surpasses state-of-the-art learning-based and zero-shot-based detectors.

*Reviewers’ Common Concerns and How We Address During Rebuttal:*

1. Question on the effectiveness of our diversely generated dataset versus other public datasets (new experiment)

- We train L2R on another major open-source diverse dataset, MAGE, and show that L2R trained on our data is more generalizable when OOD.

2. Question on the effectiveness of our OOD evaluation (clarification)

- We further detail the characteristics of our training dataset compared with the OOD dataset in Table 2 and show how they differ in data generation models, text length, decoding strategy, and domains.

3. Lack of ablation study on calibration loss and data diversity (presentation)

- We move the ablation studies from the appendix (already included in the initial submission) to the main paper.


*We believe that L2R takes a gradient step in a promising direction to improve the generalizability of LLM-generated text detection.*

---

### Meta-Review · Area_Chair_h56N · 2024-12-22

**Metareview:**

Summary:

This paper introduces a new framework called Learning2Rewrite to detect AI-generated text that generalizes to unseen domains. It leverages the insight that LLMs inherently modify AI-generated content less than human-written text when tasked with rewriting.  It shows that training LLMs to minimize alterations on AI-generated inputs, amplifies this disparity, and yields a more distinguishable and generalizable edit distance across diverse text distributions. It conducts experiments on data from 21 independent domains and four major LLMs, and finds that reinforcing LLMs’ inherent rewriting tendencies offers a robust and scalable solution for detecting AI-generated text.

Strengths:

The problem of detection of AI-generated text is very important, and the paper using LLMs’ rewriting behavior as a detection mechanism brings fresh angles and methodological innovations to approach the problem.

Weaknesses

Reviewer DZDQ raised many valid concerns regarding the original submission, such as fair comparison with baselines, impact of different decoding strategies, evaluations with OOD datasets created in different ways, etc. The authors did many experiments to address these concerns.

Reviewer 8px3 still thinks why and how fine-tuning on the proposed data set can improve the OOD performance are not clear after the first-round discussion.

I tend to agree with Reviewer DZDQ that the paper could benefit from another round of revision, given the large amount of new experiments conducted in the rebuttal phase.

**Additional Comments On Reviewer Discussion:**

Reviewer NyWU raised concerns on lack of an ablation study and usefulness of fine-tuning, for which the authors added new experiments during the discussion period.

---

### Decision · Program_Chairs · 2025-01-22

Reject